# Ethnic variations in cardiovascular disease (CVD) risk factors and associations with prevalent CVD and CVD mortality in the United States

**Queenie Cheung**[1], **Sean Wharton**[1,2], **Andrea Josse**[1], **Jennifer L. Kuk**[1]*

**1** School of Kinesiology and Health Science, York University, Toronto, Ontario, Canada, **2** The Wharton Medical Clinic, Hamilton, Ontario, Canada

* jennkuk@yorku.ca

## Abstract

### Objective

To explore the association between ethnicity and cardiovascular disease (CVD) risk factors, including physical inactivity, obesity, hypertension, type 2 diabetes (T2D), lack of health insurance and low family income in a nationally representative sample of U.S. adults.

### Research design and methods

Adults from the National Health and Nutrition Examination Survey (NHANES 2011-2020, n = 17,355) were classified as having CVD risk factors based on both self-reported and metabolic data. Ethnic differences in how these CVD risk factors relate to prevalent CVD and CVD mortality was examined in Whites, Blacks, Asians and Hispanics.

### Results

Compared to Whites, significant disparities were noted in several CVD risk factors in ethnic minorities, such as lower PA, lower income, and more prevalent metabolic risk factors. Blacks and Hispanics commonly had higher prevalent CVD risk as compared to Whites even after adjusting for income and metabolic risk factors. Physical inactivity was most strongly associated with prevalent CVD and CVD mortality among Whites and Blacks. There were no ethnic differences in the inverse association between income and prevalent CVD risk, but Blacks with low income were associated with the greatest elevated CVD mortality. Hypertension and T2D were similarly related with prevalent CVD across ethnic groups, but Blacks and Hispanics with hypertension or T2D were at greater CVD mortality risk as compared to Whites.

### Conclusion

Our study identified that socioeconomic and metabolic risk factors may relate differently to CVD outcomes among ethnic minority groups in the United States. Addressing these ethnic disparities in health warrants further investigation.

**Data availability statement:** NHANES is publicly available online at https://wwwn.cdc.gov/nchs/nhanes/Default.aspx

**Funding:** The author(s) received no specific funding for this work.

**Competing interests:** The authors have declared that no competing interests exist.

## Introduction

Cardiovascular disease (CVD) remains a leading cause of morbidity and mortality worldwide, with significant public health implications [1–5]. In the United States, CVD burden varies significantly across ethnic groups, leading to pronounced disparities in prevalence and mortality rates [6–12]. CVD risk factors include modifiable and non-modifiable risk factors [13]. Common modifiable risk factors include high blood pressure, smoking, diabetes, high cholesterol, physical inactivity, alcohol and obesity [13]. Understanding how these modifiable risk factors contribute to disparities in ethnic differences in CVD is essential for developing targeted interventions and policies to address the impact of CVD on diverse populations.

Ethnic differences in the prevalence of risk factors such as physical inactivity, type 2 diabetes (T2D), hypertension, low family income, obesity, and healthcare access are well-documented [14]. However, how these risk factors may differ by ethnicity in their association with CVD prevalence and mortality is less explored. Ethnic minorities in the United States, including Blacks, Asians and Hispanics often face unique barriers to physical activity due to socioeconomic and environmental constraints [15–18]. Similarly, the prevalence of T2D and hypertension varies by ethnicity [19–23], and is also influenced by lifestyle, and healthcare access [24–26]. Socioeconomic status is a critical health determinant, often associated with limited access to nutritious foods, and quality healthcare [24,25,27,28]. Obesity, a major CVD risk factor, is more prevalent in certain ethnic groups, and risk for obesity can be exacerbated by socioeconomic and cultural factors [29,30]. Ethnic disparities in healthcare access further increase CVD risk, especially in underserved populations [2,24,29].

This manuscript aims to address this gap by examining how common CVD risk factors (physical inactivity, low family income, lack of health insurance, hypertension, T2D and obesity) may differ in prevalence and their associations with prevent CVD and CVD mortality outcomes in White, Black, Asian and Hispanic adults in the United States.

## Methods and procedures

### Survey participants

This study is a cross-sectional study using data from the National Health and Nutritional Examination Survey (NHANES) 2011-2020 as these are the years that Asian ethnicity were recorded. The NHANES is a nationally representative survey conducted by the Centers for Disease Control and Prevention (CDC) in the United States [31]. All study participants gave their informed written consent before participation in the examination, and the study protocol was approved by the National Center for Health Statistics [31]. Public-use data files were used and thus this study did not require further ethical review from York University's Research Ethics board. The dataset initially included 39,156 individuals. Participants under the age of 40 years and over the age of 80 years were excluded to focus on those with a higher likelihood of experiencing lifestyle associated CVD risk (n = 15,066). Participants with missing data for variables of interest (CVD, physical activity, health insurance, body mass index (BMI), blood pressure, diabetes, family income) or a BMI below 18.5 kg/m² (n = 6735), or those in the "other" ethnic group category (n = 1996) were excluded from the analytical dataset leaving 17,355 individuals. All analyses were performed in accordance with relevant NHANES analytical guidelines [32].

### Survey methods

Age, sex, ethnicity, poverty-income ratio, physical activity status, access to health insurance were obtained during the interview portion of the survey by trained personnel [33]. Participants were asked to self-identify as: Mexican American, Other Hispanic, Non-Hispanic

White, Non-Hispanic Black, Non-Hispanic Asian or Other – including multi-racial [34]. For this study, Mexican American and Other Hispanic were collapsed into a single Hispanic group due to low sample size. Those in the Other ethnic group were excluded due to low sample size. Participants were asked the amount of time they typically spent doing moderate or vigorous sports, fitness or recreational activities or at work [35]. Individuals were considered physically active if the sum of the reported durations for vigorous- or moderate-intensity leisure or work ≥ 150 minutes per week [35].

During the NHANES examination, height was measured using a stadiometer, while weight was measured using a digital scale [34]. BMI was calculated as weight (in kilograms) divided by height (in meters) squared. Individuals were classified as having obesity using the CDC cut-offs: BMI ≥ 30.0 kg/m² [36], and Asian specific BMI categories as defined by the World Health Organization (WHO): BMI ≥ 25.0 kg/m² [37]. Those considered underweight (BMI below 18.5 kg/m²) were excluded as the causes of CVD are likely altered in underweight [38], and due to low sample size.

Family income was defined as the Family Poverty Income Ratio (PIR), is a measure of income relative to the federal poverty guidelines as outlined in the NHANES analytical guidelines [33]. Individuals were classified as having health insurance if they answered "yes" to the question: "Are you covered by health insurance or some other kind of health care plan?" [35].

Fasting metabolic blood data were collected through standardized procedures [35]. Individuals were classified as hypertensive if they were taking antihypertensive medication, had a measured systolic blood pressure >= 140 mm Hg, had a diastolic blood pressure >= 90 mm Hg during the clinical exam, or had a physician diagnosis of hypertension [39]. Individuals were classified as having T2D if they had a fasting glucose > 7 mmol/L, HbA1c > 6.5%, use of diabetes medication or had a physician diagnosis of diabetes [40].

## Prevalent CVD and CVD mortality

Participants who answered, 'yes' to the question: "Has a doctor or other health professional ever told you that you had a myocardial infarction, coronary heart disease, or stroke?" were considered to have prevalent cardiovascular disease [41]. The National Center for Health Statistics merged data from death certificates and NHANES records until December 31, 2019 [42]. CVD death was defined as ICD-10 codes I00-I09, I11, I13, and I20-I51 [42].

## Statistical analysis

Participant characteristics are reported as prevalence ± standard error (SE) and means ± SE. Ethnic differences in the odds of prevalent CVD associated with common CVD risk factors (physical inactivity, low family income, lack of health insurance, hypertension, T2D and obesity) were examined using logistic regression. Ethnic differences in CVD mortality risk associated with each CVD risk factor was assessed using Fine and Gray's sub-distribution Cox Proportional hazard model that considers competing risks for all-other non-CVD causes of mortality [43]. Both analyses examined ethnicity and CVD risk factor main effects and interactions adjusted for sex and age. All analyses were weighted to be representative of the U.S. population and performed using SAS version 9.4 (SAS Institute, Cary, NC) with statistical significance established at $p \leq 0.05$.

## Results

Table 1 shows participant characteristics stratified by ethnicity. In general, the rates of CVD risk factors (i.e., physical inactivity levels, lack of health insurance, low-income rates, T2D, hypertension and obesity) were higher among ethnic minority groups with some exceptions.

Asians and Whites were not different in terms of health insurance and low income, and Asians and Hispanics were less likely to have hypertension and obesity than Whites.

There was a significant physical activity and ethnicity main effect (p < 0.05), with a potential ethnicity by physical activity interaction (prevalent CVD p = 0.07, CVD mortality p < 0.0001, Fig 1). Insufficient physical activity was associated with and increased odds of prevalent CVD and CVD mortality. In the post hoc analysis, physical inactivity was associated with a greater odds of prevalent CVD for Black and White individuals, but not Hispanic and Asian individuals. Similarly, for CVD mortality, there was a significant ethnicity by physical activity interaction (p < 0.0001, Fig 1), indicating a stronger association between physical activity and CVD mortality in White individuals, than other ethnicities.

Ethnic differences in the association between income and health insurance was more clearly observed with CVD mortality as compared to prevalent CVD (Fig 2). For CVD mortality there was a significant interaction of ethnicity by family income (p < 0.0001) and lack of health insurance (p < 0.0001), with low income and lack of health insurance being particularly strongly associated with CVD mortality in White and Black adults.

Ethnic differences in the associations between metabolic risk factors and prevalent CVD and CVD mortality is presented in Fig 3. There were ethnic by T2D and hypertension interaction effects in that Black and Hispanic individuals with T2D or hypertension were at substantially higher CVD mortality risk than White. However, Black individuals even without T2D or hypertension also tended to have an elevated CVD mortality risk, though this did not reach statistical significance (Hypertension HR = 3.1, 95% CI (0.7-4.9); T2D HR = 2.1, 95% CI (0.9-3.3). There was a main effect of obesity on prevalent CVD, with White, Black and Hispanic individuals at higher odds of prevalent CVD (p < 0.05). For CVD mortality, obesity was only associated with significantly higher CVD mortality risk in Whites (p < 0.05).

## Discussion

Our study identified differences in the associations between CVD risk factors and CVD outcomes between ethnicities, highlighting how ethnicity may influence cardiovascular health

**Table 1. Baseline characteristics stratified by ethnic group (NHANES 2011-2020).**

|  | White | Black | Asian | Hispanics |
|---|---|---|---|---|
| **N** | 7218 | 4229 | 2362 | 4378 |
| **Age, years** | 59.1 ± 0.2 | 56.2 ± 0.3* | 55.7 ± 0.4* | 54.3 ± 0.3 |
| **Male, %** | 48.6 ± 0.5 | 44.4 ± 0.6* | 46.5 ± 0.8 | 49.9 ± 0.6* |
| **Active, %** | 71.6 ± 0.7 | 63.7 ± 0.9* | 61.2 ± 1.4* | 64.9 ± 1.1* |
| **Health Insurance, %** | 89.4 ± 0.8 | 79.1 ± 0.9* | 87.3 ± 1.0 | 64.2 ± 1.4* |
| **Low Income, %** | 42.9 ± 1.7 | 68.4 ± 1.8* | 45.2 ± 2.4 | 74.1 ± 1.5* |
| **Hypertension, %** | 36.9 ± 0.9 | 42.9 ± 0.8* | 28.5 ± 1.1* | 24.2 ± 1.1* |
| **T2D, %** | 13.8 ± 0.5 | 18.8 ± 0.7* | 16.2 ± 1.0* | 16.3 ± 0.8 |
| **Obesity, %** | 38.6 ± 0.9 | 48.9 ± 1.0* | 30.8 ± 1.2* | 45.7 ± 1.0* |
| **CVD, %** | 7.9 ± 0.4 | 6.9 ± 0.3* | 3.4 ± 0.4* | 4.1 ± 0.4* |
| **All-cause mortality, %** | 3.9 ± 0.2 | 3.2 ± 0.3* | 1.3 ± 0.3* | 1.6 ± 0.2* |
| **CVD Mortality, %** | 1.5 ± 0.2 | 1.6 ± 0.2 | 0.4 ± 0.1* | 0.7 ± 0.1* |
| **Follow up, years** | 4.9 ± 0.1 | 4.8 ± 0.2 | 4.8 ± 0.2 | 4.9 ± 0.2 |

Values presented as means or prevalence ± standard error.

T2D = Type 2 diabetes; CVD = Cardiovascular Disease (Stroke, Myocardial Infarction, Coronary Heart Disease).

*Significant difference (p < 0.05) compared to White.

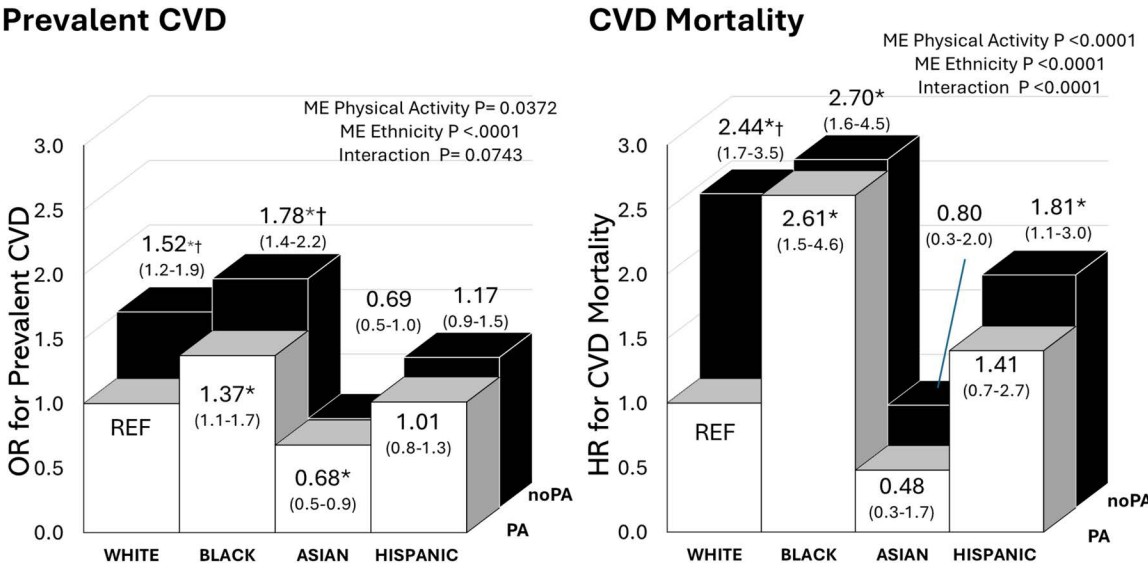

**Fig 1. Association of physical activity (PA) and ethnicity with prevalent CVD and CVD mortality.** noPA= Physically Inactive (< 150 min/week). ME = main effect. Models were adjusted for age and sex. * significant difference from White Physically Active (REF) (P < 0.05). † significant difference within each ethnic group.

risk. Certain CVD risk factors such as hypertension were more common in Blacks, and were also more strongly associated with prevalent CVD and CVD mortality, while physical activity was less strongly associated with CVD in ethnic minorities as compared to White adults. Persistent health disparities among certain ethnic minorities in CVD prevalence and mortality may be compounded by socioeconomic factors that may limit healthcare access and result in a higher burden of traditional CVD risk factors [44–47]. Further research is needed to determine why these differences exist and how health policy and healthcare can be improved to address these disparities effectively.

While ethnicity itself does not directly cause CVD, certain groups exhibit higher susceptibility and prevalence rates for conditions like hypertension, diabetes, and coronary artery disease due to genetic, environmental, and socioeconomic factors [12,21,44–48]. Our study, consistent with current literature, found that Black and Hispanic individuals carry a heavier burden of traditional CVD risk factors such as obesity, hypertension and diabetes [8,26,49–51]. This could be related to greater socioeconomic barriers, including lower income and education levels, which may limit access to healthcare and resources promoting healthy lifestyles, contributing to higher rates of CVD risk factors and potentially delayed diagnosis and treatment [44–46,52]. Ethnic-specific differences in how risk factors influence cardiovascular health [53], may contribute to ethnic differences in CVD hard outcomes such as heart attack, stroke, and coronary heart disease and CVD mortality.

Studies reveal that individuals with T2D and hypertension exhibit significantly elevated prevalent CVD and CVD mortality rates compared to those without [39,54]. We and others in the literature, show that ethnic minorities have a higher prevalence of T2D and hypertension that contribute to a higher risk for CVD morbidity and mortality [39,55,56]. Black adults under the age of 65 years are thought to face heightened CVD mortality risks due to early onset of T2D, hypertension and lower socioeconomic status [49]. Moreover, hypertension control appears less effective among Black individuals than among White individuals, primarily attributed to challenges with medication affordability [7]. We observe that

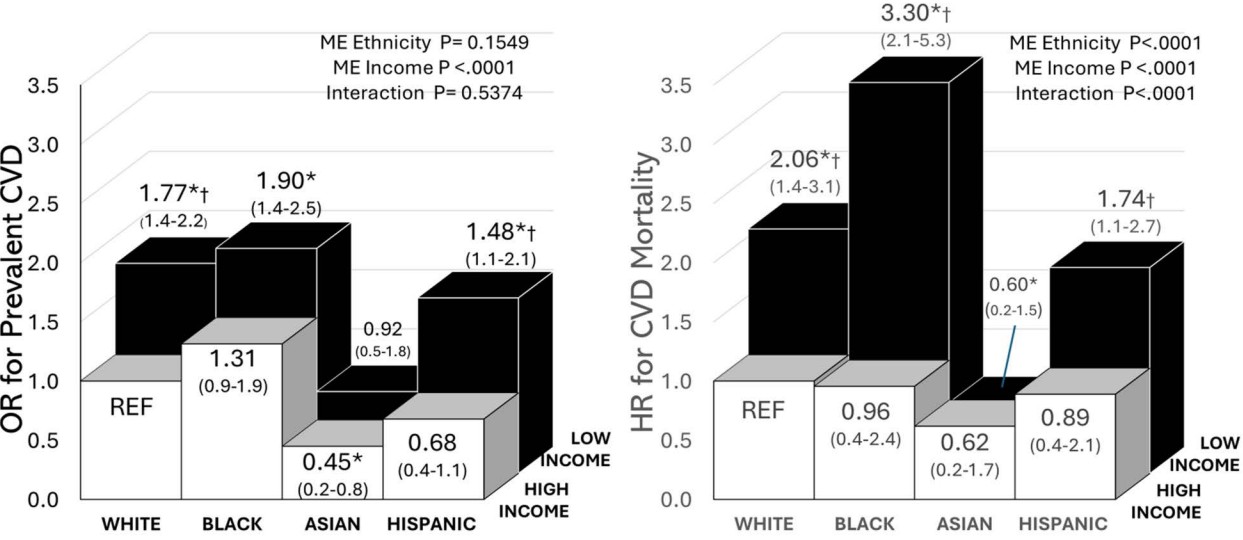

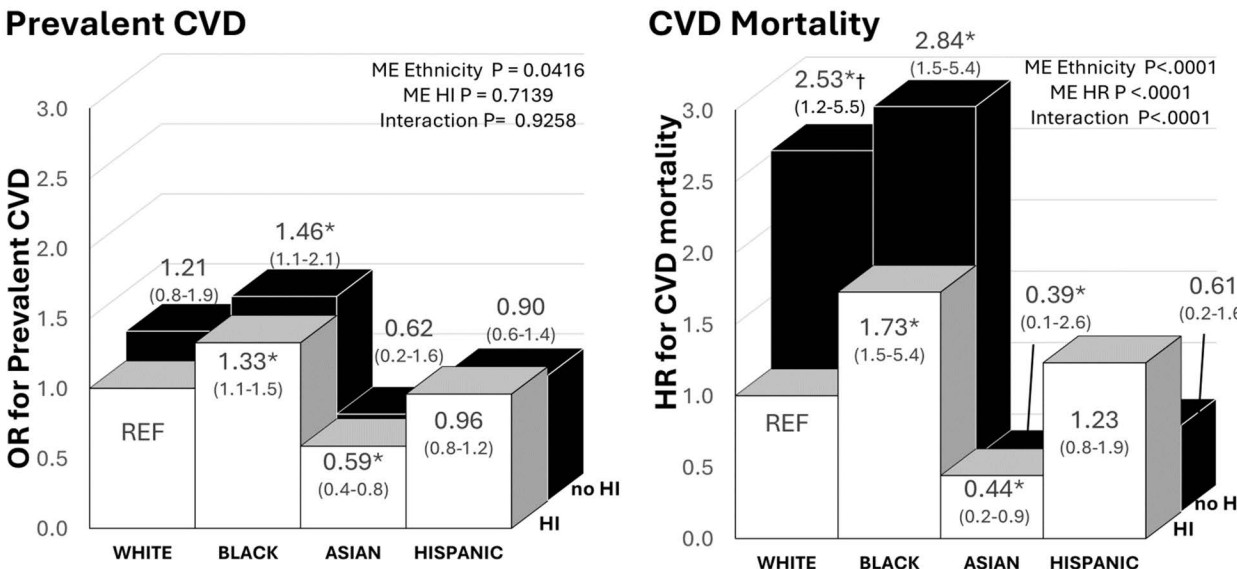

**Fig 2. Association of socioeconomic risk factors (income & health insurance) and ethnicity with prevalent CVD and mortality.** Low Income (Family PIR <3). HI = Healthcare Insurance. Main Effect = ME. Models were adjusted for age and sex. * significant difference from White High Income or HI (REF) (P < 0.05). † significant difference within each ethnic group.

Black individuals face heightened risks of CVD mortality, and Hispanics were found to have increased risk of both prevalent CVD and CVD mortality. Even without hypertension or diabetes, Black individuals face an increased risk of prevalent CVD and CVD mortality compared to their counterparts with these comorbidities. In summary, while ethnic disparities in hypertension and T2D risk vary, these conditions collectively contribute to elevated prevalent CVD risk and CVD mortality among ethnic minorities.

Socioeconomic factors are well known to be associated with health and mortality outcomes [28]. Lower family income is linked with a higher prevalence of T2D, hypertension, and

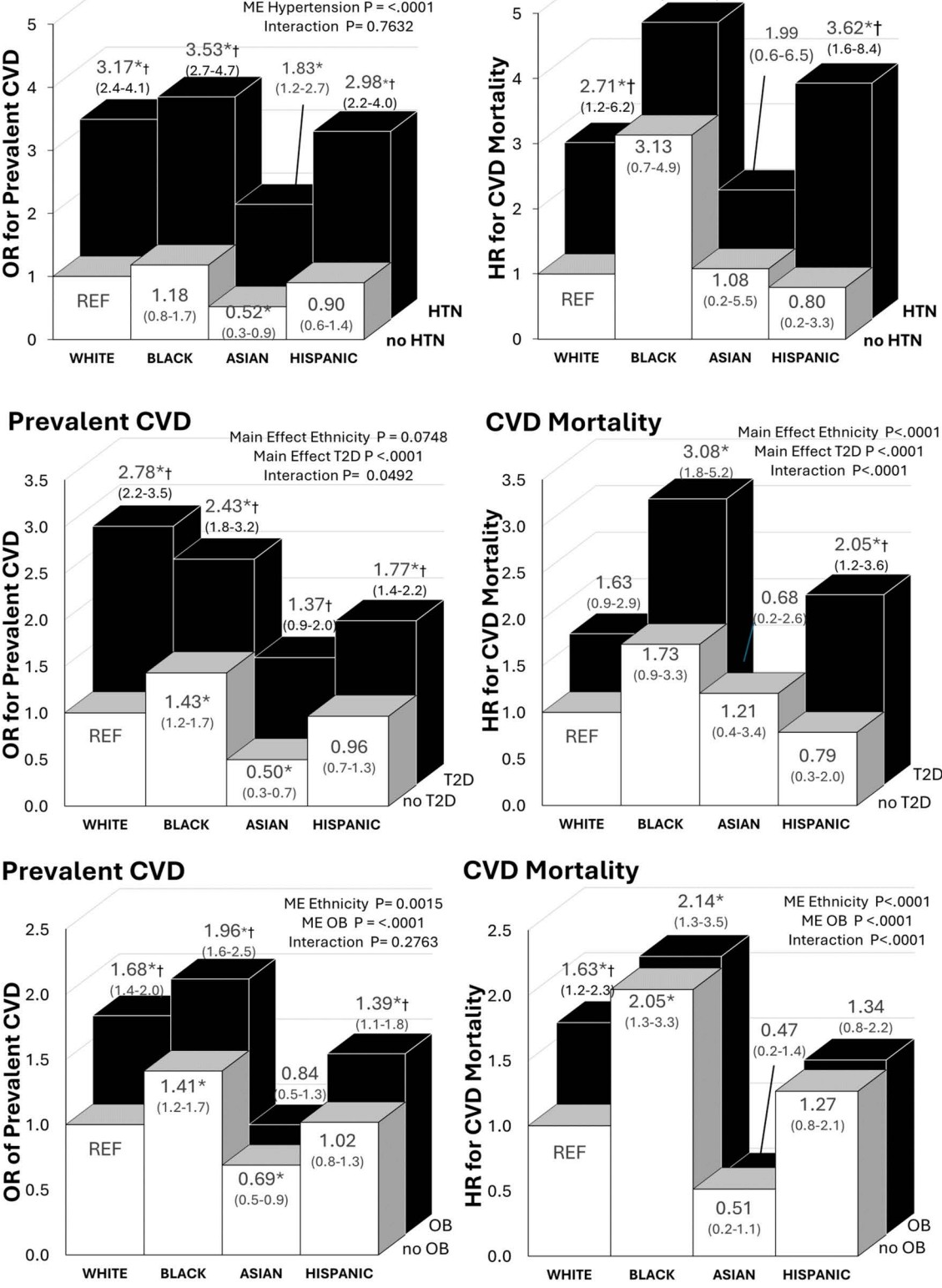

**Fig 3. Association of metabolic risk factors (T2D, hypertension and obesity) and ethnicity with prevalent CVD and mortality.**
HTN = hypertension. T2D = Type II Diabetes. OB = Obesity (BMI>=30kg/m²) Asian individuals (BMI>=27 kg/m²). ME= Main

Effect. Models were adjusted for age and sex. * significant difference from White Healthy (REF) (P < 0.05). † significant difference within each ethnic group (P < 0.05).

coronary artery disease [25,28,45]. Interestingly, we did not observe an association between low family income and CVD in Asians. This may be in part due to differences in cultural practices that might offset some of the risks associated with low income [25,27]. In Asian cultures, living in a multigenerational home is more common and may allow for a more supportive arrangement, where chores such as meal preparation and financial burdens may be shared [57]. The presence of caretakers at home alleviate may some financial and emotional burdens and ensures individuals may get diagnosed earlier and those with illness may be more likely to get health care [57]. This contrasts with Western cultural norms where older generations often reside in caretaker homes or live independently, leading to a different dynamic in familial support and financial responsibilities [57]. However, Hispanic and Black individuals have a similar prevalence of living in multi-generation homes as compared to Asians [58]. Thus, the reasons for these differences require further investigation, but may be associated with differences in health care. Black and Hispanic individuals encounter longer wait times and delays in medical care, systemic biases, and limited healthcare access that can be further compounded by low income (26) which may result in the worse CVD outcomes observed.

Physical activity confers a range of health benefits, including improved weight management, and reduced risks of metabolic and chronic diseases such as CVD [59,60]. However, there are differences in how physical activity relates with health outcomes depending on whether physical activity is done in their leisure time or occupationally [61–65]. While leisure time physical activity generally correlates positively with health outcomes, the relationship between occupational physical activity and health is more controversial [61–63,65,66]. From a physiological standpoint, physical activity, whether in leisure or occupational contexts, should confer the same health benefits. However, job-related stress, occupational hazards, or the low intensity of most occupational physical activity may explain why occupational activity is often associated with less positive or even negative health outcomes [61–63,65,66]. Alternatively, it is plausible that the positive characteristics that enable individuals to be physically active in their leisure time (e.g., sufficient income, manageable stress, sufficient time, conducive environment, etc.) also contribute to the observed health benefits among those who are leisurely active [16,17,63,67]. Among Blacks and Hispanics, 66% report some form of leisure time physical activity, which is lower compared to the 78% leisure time physical activity engagement observed in Whites [17]. Conversely, Hispanics tend to engage in the highest prevalence of occupational physical activity among ethnic groups [61–63]. In this study, our measure of physical activity allows for both domains of physical activity in accordance with current physical activity guidelines [60]. As expected, we observe that physically inactive White and Black individuals showed higher odds of prevalent CVD compared to their active counterparts. However, physical activity was only related with CVD mortality in White individuals. In fact, there was no difference in mortality risk between active and inactive individuals in those of Black, Asian or Hispanic ethnicity. This may suggest that physical activity may be more strongly related with CVD among Whites compared to other ethnic groups. Our findings underscore the need for tailored strategies to enhance cardiovascular health outcomes, particularly among ethnic minorities, where traditional measures may not fully capture protective effects.

There are several strengths and limitations worth mentioning. This study utilized a large, representative NHANES dataset offering insights into CVD risk factors, prevalent CVD, and mortality risk in four ethnic groups in the U.S. population [32,41]. However, its

cross-sectional design limits establishing causality, and reliance on self-reported data for prevalence CVD, physical activities and medications may introduce reporting biases. Language barriers or lack of healthcare access may also have resulted in missed diagnoses of CVD, particularly in certain ethnic minority groups, as the study used self-report doctor diagnosis [68]. Further, this study does not account for within-group heterogeneity for factors such as socioeconomic status among individuals of the same ethnicity, which may oversimplify the findings. Errors in the assessment of factors may contribute to biases to the null. Further, there may be complex interaction effects between the factors that the current analysis was unable to address. Combining all Asian and Hispanic subgroups into a single category may obscure potential differences in CVD risk factors and outcomes among distinct ethnicities [69]. For example, South Asians generally exhibit elevated CVD risk due to higher rates of insulin resistance, diabetes, central obesity, and atherogenic dyslipidemia than White individuals, whereas East Asians have been demonstrated to have a lower likelihood of dying from a CVD event [11,70].

Targeted public health campaigns are necessary to address CVD risk factors and improve prevention strategies. This study provides evidence that traditional CVD risk factors may relate with CVD mortality different in ethnic minority groups. Future research should delve deeper into the reasons behind the varying associations between CVD risks and CVD mortality to develop targeted public health strategies that address the specific needs of each population and promote health equity.

## Supporting information

**S1 Checklist. PLOS One human subjects research checklist.**
(DOCX)

## Author contributions

**Conceptualization:** Queenie Cheung, Sean Wharton, Andrea Josse, Jennifer L. Kuk.

**Formal analysis:** Queenie Cheung, Jennifer L. Kuk.

**Methodology:** Queenie Cheung, Sean Wharton, Andrea Josse, Jennifer L. Kuk.

**Supervision:** Andrea Josse, Jennifer L. Kuk.

**Writing – original draft:** Queenie Cheung.

**Writing – review & editing:** Sean Wharton, Andrea Josse, Jennifer L. Kuk.

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
