## [Decision Letter · Decision Letter 0]

4 Dec 2024

PONE-D-24-43621Ethnic Variations in Cardiovascular Disease (CVD) Risk Factors and Associations with Prevalent CVD and CVD MortalityPLOS ONE

Dear Dr. Kuk,

Thank you for submitting your manuscript to PLOS ONE. After careful consideration, we feel that it has merit but does not fully meet PLOS ONE’s publication criteria as it currently stands. Therefore, we invite you to submit a revised version of the manuscript that addresses the points raised during the review process.

After careful review, I acknowledge its merit; however, there are several considerations that need to be addressed before a final decision can be made. Kindly attend to the recommendations provided by Reviewer 1.

We look forward to receiving your revised manuscript.

Kind regards,

Neftali Eduardo Antonio-Villa, MD PhD

Academic Editor

PLOS ONE

Journal Requirements:

2. We noted in your submission details that a portion of your manuscript may have been presented or published elsewhere. [A version of this work will be published as part of Queenie Cheung’s masters thesis (first author) in the York University Electronic Theses and Dissertations Collection (https://hdl.handle.net/10315/26310).  This is not peer reviewed and the current submitted version has been edited.] Please clarify whether this [conference proceeding or publication] was peer-reviewed and formally published. If this work was previously peer-reviewed and published, in the cover letter please provide the reason that this work does not constitute dual publication and should be included in the current manuscript.

Reviewers' comments:

Reviewer's Responses to Questions

**Comments to the Author**

1. Is the manuscript technically sound, and do the data support the conclusions?

Reviewer #1: Partly

Reviewer #2: Yes

2. Has the statistical analysis been performed appropriately and rigorously? 

Reviewer #1: Yes

Reviewer #2: Yes

3. Have the authors made all data underlying the findings in their manuscript fully available?

Reviewer #1: Yes

Reviewer #2: Yes

4. Is the manuscript presented in an intelligible fashion and written in standard English?

Reviewer #1: Yes

Reviewer #2: Yes

5. Review Comments to the Author

Reviewer #1: Dear Authors,

Thank you for submitting your paper and the opportunity to review your valuable manuscript. I have several suggestions that I hope it helps.

Introduction:

1. The whole introduction section is presented as a single paragraph. It might benefit from being organized into multiple paragraphs to enhance its readability and logical flow. For example, as a suggestion, consider building a background in the first paragraph, specifying the research question in the second, providing more data and areas for research in the third, and stating the aim and objective of the study in the final paragraph.

2. Throughout the introduction and the manuscript, you frequently refer to "CVD risk factors.” It would be helpful to explicitly state what these risk factors are and provide specific reference studies/texts that identify these as established risk factors for CVD.

3. The term "ethnic minorities" is quite broad and can vary significantly even within different regions of a city. Please specify the ethnic groups and regions that your research focuses on, providing background on any previous research in this area if available.

4. To make the research question and aim more precise, please clarify which ethnic groups and regions are being studied, what specific risk factors are being examined, and what CVD outcomes of interest are included (such as mortality, morbidity, or incidence).

5. Using the same reference multiple times for various claims, such as reference (1) in the introduction, may raise concerns about the depth of the literature review. Please consider diversifying your references to provide a more comprehensive background.

Methods:

1. The study excludes participants under 40 and over 80 years old. The rationale for selecting these specific age thresholds is not adequately justified. Please provide a clear justification for this exclusion criterion with reliable sources. This is important because excluding these age groups may potentially affect the generalizability of your findings by omitting populations who are at risk of CVD, introducing potential selection bias.

2. Similar to the age exclusion, excluding individuals with a BMI below 18.5 requires further explanation. Please elaborate on the reasoning behind this criterion and clarify why underweight individuals are excluded. Additionally, if this exclusion is necessary, explain why it is not applied to groups with higher BMI values.

3. Please provide a rationale for excluding individuals categorized under "Other" ethnic groups. Excluding these participants may result in the loss of valuable data on minorities within minorities, potentially affecting the inclusivity and representativeness of your study by omitting those who may be at higher risk due to smaller population sizes.

4. You might consider discussing the decision to combine "Mexican American" and "Other Hispanic" groups. Some may suggest noting in the limitations section that this aggregation could mask important differences in CVD risk factors by pooling ethnic groups that are not necessarily identical.

5. The definition of prevalent CVD is based on participants' self-reports. Self-reported diagnoses can be unreliable due to recall errors or lack of medical awareness, and these issues may be more prevalent in certain ethnic groups due to language barriers. You may want to address this limitation in your methodology.

Results:

1. The Results section contains some vague statements, such as "with some exceptions.” It would be beneficial to avoid such terms in the paper, specifically the Results section, and instead provide specific details. For example, clarify which risk factors were higher among ethnic minority groups, what the exceptions were, and specify the ethnic groups involved.

2. In the sentence mentioning "a potential ethnicity by physical activity interaction (p=0.07)”, it's unclear whether this p-value indicates statistical significance, as it is above the conventional threshold of 0.05. Please clarify the interpretation of this result.

3. By not accounting for within-group sources of heterogeneity, such as differences in socioeconomic status among individuals of the same ethnicity, the results may oversimplify complex relationships. You might consider addressing this potential limitation in the Discussion section.

Discussion:

1. The Discussion section would benefit from a clear and specific conclusion that summarizes the main findings of the study and their implications, highlighting the key takeaways.

2. Also, it would be helpful to offer a clear, actionable clinical note or practical recommendations for clinicians, policymakers, public health practitioners, etc., based on your findings.

3. The section on limitations is brief and lacks depth. While some limitations are mentioned (e.g., cross-sectional design, self-reported data), they are not thoroughly explored. Expanding on the limitations will help readers understand the context and potential constraints of your study.

Reviewer #2: Although well thought out and executed, after reviewing the literature, I did not find the findings to be unique. Additionally, physical activity is based on recall and hence, basing results on this may not be accurate

6. PLOS authors have the option to publish the peer review history of their article (what does this mean? ). If published, this will include your full peer review and any attached files.

**Do you want your identity to be public for this peer review?** For information about this choice, including consent withdrawal, please see our Privacy Policy .

Reviewer #1: **Yes: ** Afshin Heidari

Reviewer #2: No

---

## [Author Response · Author response to Decision Letter 1]

20 Jan 2025

Thank you for the chance to revise and improve our manuscript. We have provided a point-by-point response for each of the comments below and a revised version with the changes highlighted. The formatting of the title and manuscript have been revised to comply with guidelines. The current manuscript is a revised version of a thesis that was placed in an open repository. The thesis was not peer reviewed and would be more akin to placing a draft version in a pre-print archive.

Comments to the Author

Reviewer #1:

Comment 1 - Introduction:

1. The whole introduction section is presented as a single paragraph. It might benefit from being organized into multiple paragraphs to enhance its readability and logical flow. For example, as a suggestion, consider building a background in the first paragraph, specifying the research question in the second, providing more data and areas for research in the third, and stating the aim and objective of the study in the final paragraph.

Response � Thank you, we have re-organized the introduction in multiple paragraphs and have hopefully enhanced the logic flow (pages 1-2).

Comment 2. Throughout the introduction and the manuscript, you frequently refer to "CVD risk factors.” It would be helpful to explicitly state what these risk factors are and provide specific reference studies/texts that identify these as established risk factors for CVD.

Response � We now name to the specific risk factors we examine at the start of the introduction (1st paragraph). Because we wanted to examine commonly accepted CVD risk factors, we opted to use the factors listed from the Heart and stroke instead of a single study.

Comment 3. The term "ethnic minorities" is quite broad and can vary significantly even within different regions of a city. Please specify the ethnic groups and regions that your research focuses on, providing background on any previous research in this area if available.

Response �We have revised the title, abstract and introduction to indicate the study is examining ethnic minorities in the United States. Our analyses using NHANES is weighted to be nationally representative (3rd line of the methods section):

“The NHANES is a nationally representative survey conducted by the Centers for Disease Control and Prevention (CDC) in the United States” (top of page 3, line 3-4)

Comment 4. To make the research question and aim more precise, please clarify which ethnic groups and regions are being studied, what specific risk factors are being examined, and what CVD outcomes of interest are included (such as mortality, morbidity, or incidence).

Response � We have added the requested details to the aim at the end of the introduction (page 2).

“This manuscript aims to address this gap by examining how common CVD risk factors (physical inactivity, low family income, lack of health insurance, hypertension, T2D and obesity) may differ in prevalence and their associations with prevent CVD and CVD mortality outcomes in White, Black, Asian and Hispanic adults in the United States. “

Comment 5. Using the same reference multiple times for various claims, such as reference (1) in the introduction, may raise concerns about the depth of the literature review. Please consider diversifying your references to provide a more comprehensive background.

Response � We have now included more references throughout the paper.

Methods:

Comment 1. The study excludes participants under 40 and over 80 years old. The rationale for selecting these specific age thresholds is not adequately justified. Please provide a clear justification for this exclusion criterion with reliable sources. This is important because excluding these age groups may potentially affect the generalizability of your findings by omitting populations who are at risk of CVD, introducing potential selection bias.

Response � The current study is examining ethnic differences in how CVD risk factors relate with prevalent CVD and CVD mortality. Because CVD is a chronic disease, focusing on middle aged adults will allow time for those risk factors to develop and result in CVD (1), (2-3) We chose to exclude those over the ages of 80 due to concerns about competing risks from other causes of mortality (3), which would obscure our ability to see associations between the risk factors and CVD outcomes (4).

2. Similar to the age exclusion, excluding individuals with a BMI below 18.5 requires further explanation. Please elaborate on the reasoning behind this criterion and clarify why underweight individuals are excluded. Additionally, if this exclusion is necessary, explain why it is not applied to groups with higher BMI values.

We chose to exclude individuals with a BMI less than 18.5 due to sample size and mechanistic differences in the causes of CVD in underweight versus other BMI categories (4). Thus, the relationship between the examined CVD risk factors and CVD are likely altered in those with underweight, but we are underpowered to examine this.

“Those considered underweight (BMI below 18.5 kg/m2) were excluded as the causes of CVD are likely altered in underweight (38), and due to low sample size.” – Page 4, paragraph 2.

Comment 3. Please provide a rationale for excluding individuals categorized under "Other" ethnic groups. Excluding these participants may result in the loss of valuable data on minorities within minorities, potentially affecting the inclusivity and representativeness of your study by omitting those who may be at higher risk due to smaller population sizes.

Response �We agree that exclusion of the ‘other’ ethnic group is unfortunate. However, we were limited by sample size which resulting in cell sizes of less than 5 for some of our models (i.e. ethnicity by CVD risk factor by CVD mortality) that created model instability and wide confidence intervals. Because of the heterogeneity of the group, we also did not feel that it would be appropriate to combine the ‘other’ ethnic group with any other group.

“Those in the Other ethnic group were excluded due to low sample size.” (page 3-4).

Comment 4. You might consider discussing the decision to combine "Mexican American" and "Other Hispanic" groups. Some may suggest noting in the limitations section that this aggregation could mask important differences in CVD risk factors by pooling ethnic groups that are not necessarily identical.

Response � Again, we combined these groups due to low absolute number of CVD deaths in “Mexican American” (n=36) and “Other Hispanic” (n=28). When looking at general patterns of the associations in these groups, they were similar which made us more confident in combining these groups. We have also included this as part of our limitations (page 16, paragraph 2)

“Combining all Asian and Hispanic subgroups into a single category may obscure potential differences in CVD risk factors and outcomes among distinct ethnicities”

5. The definition of prevalent CVD is based on participants' self-reports. Self-reported diagnoses can be unreliable due to recall errors or lack of medical awareness, and these issues may be more prevalent in certain ethnic groups due to language barriers. You may want to address this limitation in your methodology.

We have included this point with a relevant reference in the limitations (page 16, paragraph 2).

“Language barriers or lack of healthcare access may also have resulted in missed diagnoses of CVD, particularly in certain ethnic minority groups, as the study used self-report doctor diagnosis (5).“

Results:

Comment 1. The Results section contains some vague statements, such as "with some exceptions.” It would be beneficial to avoid such terms in the paper, specifically the Results section, and instead provide specific details. For example, clarify which risk factors were higher among ethnic minority groups, what the exceptions were, and specify the ethnic groups involved.

Response � The exceptions are now stated in the results. (page 6).

“….with some exceptions. Asians and Whites were not different in terms of health insurance and low income, and Asians and Hispanics were less likely to have hypertension and obesity than Whites.”

2. In the sentence mentioning "a potential ethnicity by physical activity interaction (p=0.07)”, it's unclear whether this p-value indicates statistical significance, as it is above the conventional threshold of 0.05. Please clarify the interpretation of this result.

A p-value of 0.05 to indicate significance is a threshold that is widely used, but is simply one value along a continuum of probabilities. For examination of interactions with multiple groups, some clinically relevant differences can be seen with a larger p-value, which is why p-values of 0.1 (even p<0.20) are used to suggest that group differences should be investigated (6). As this approach versus the strict use of P<0.05 are debated (7), and that the parallel analyses with CVD mortality was significant, we report the actual significance value and use less definitive language so that the reader can decide which interpretation they believe. We have revised the text in the results to more clearly convey what was done (page 7).

“There was a significant physical activity and ethnicity main effect (p<0.05), with a potential ethnicity by physical activity interaction (prevalent CVD p=0.07, CVD mortality p<0.0001, Figure 1). Insufficient physical activity was associated with and increased odds of prevalent CVD and CVD mortality. In the post hoc analysis, physical inactivity was associated with a greater odds of prevalent CVD for Black and White individuals, but not Hispanic and Asian individuals. “

Comment 3. By not accounting for within-group sources of heterogeneity, such as differences in socioeconomic status among individuals of the same ethnicity, the results may oversimplify complex relationships. You might consider addressing this potential limitation in the Discussion section.

Response � Thank you, we have now listed this in the limitations section. (Page 16, paragraph 2).

“Further, this study does not account for within-group heterogeneity for factors such as socioeconomic status among individuals of the same ethnicity, which may oversimplify the findings.”

Discussion:

1. The Discussion section would benefit from a clear and specific conclusion that summarizes the main findings of the study and their implications, highlighting the key takeaways.

We have added a paragraph to summarize the findings (page 13, paragraph 1)

“Our study identified differences in the associations between risk factors and CVD outcomes between ethnicities, highlighting how ethnicity may influence cardiovascular health risk. Certain CVD risk factors such as hypertension were more common in Blacks, and were also more strongly associated with prevalent CVD and CVD mortality, while physical activity was less strongly associated with CVD in ethnic minorities as compared to White adults. Persistent health disparities among certain ethnic minorities in CVD prevalence and mortality may be compounded by socioeconomic factors that may limit healthcare access and result in a higher burden of traditional CVD risk factors. Further research is needed to determine why these differences exist and how health policy and healthcare can be improved to address these disparities effectively.”

2. Also, it would be helpful to offer a clear, actionable clinical note or practical recommendations for clinicians, policymakers, public health practitioners, etc., based on your findings.

Thank you. That is an insightful suggestion, however, given the variation in the results and the uncertainty in why these differences exist, we are uncertain on what we should recommend to clinicians, policy makers and public health practitioners in order to improve health inequities. Therefore, we have concluded that more research is needed in the last line of the abstract and the discussion (page 13, paragraph 1).

“Further research is needed to determine why these differences exist and how health policy and healthcare can be improved to address these disparities effectively.”

3. The section on limitations is brief and lacks depth. While some limitations are mentioned (e.g., cross-sectional design, self-reported data), they are not thoroughly explored. Expanding on the limitations will help readers understand the context and potential constraints of your study.

We have substantially added to the limitations in the discussion (page 16, paragraph 2).

Reviewer #2: Although well thought out and executed, after reviewing the literature, I did not find the findings to be unique. Additionally, physical activity is based on recall and hence, basing results on this may not be accurate

We are unaware of any studies that have examined this issue with so many CVD risk factors with CVD mortality and prevalent CVD in a nationally representative sample with four unique ethnic groups. Most studies that have examined ethnic differences only include White with one other ethnic group. Having all of the examinations in one study reduces the effect of sampling causing variation in the results. If the reviewer is aware of studies that have examined the same question as our analyses, then we would appreciate it if you would be able to provide these references so that we can better reflect the literature.

In terms of the limitations of physical activity, we agree that there may be self-report and recall bias. Nevertheless, self-report and objective measures of physical activity (which also are associated with error and bias) are typically correlated(8) (9), and so these differences should not invalidate our observations. In fact, errors in measurement are typically associated with biases to the null.

1. Rodgers JL, Jones J, Bolleddu SI, Vanthenapalli S, Rodgers LE, Shah K, et al. Cardiovascular Risks Associated with Gender and Aging. J Cardiovasc Dev Dis. 2019 Apr 27;6(2).

2. NHS. https://www.nhs.uk/conditions/cardiovascular-disease/. 2022. Cardiovascular disease.

3. Aïdoud A, Gana W, Poitau F, Debacq C, Leroy V, Nkodo J, et al. High Prevalence of Geriatric Conditions Among Older Adults With Cardiovascular Disease. J Am Heart Assoc. 2023 Jan 17;12(2).

4. Kwon H, Yun JM, Park JH, Cho BL, Han K, Joh HK, et al. Incidence of cardiovascular disease and mortality in underweight individuals. J Cachexia Sarcopenia Muscle. 2021 Apr;12(2):331–8.

5. Herbert BM, Johnson AE, Paasche-Orlow MK, Brooks MM, Magnani JW. Disparities in Reporting a History of Cardiovascular Disease Among Adults With Limited English Proficiency and Angina. JAMA Netw Open. 2021 Dec 1;4(12):e2138780.

6. Thiese MS, Ronna B, Ott U. P value interpretations and considerations. J Thorac Dis. 2016 Sep;8(9):E928–31.

7. Feinstein AR. P-Values and Confidence Intervals: Two Sides of the Same Unsatisfactory Coin. J Clin Epidemiol. 1998 Apr;51(4):355–60.

8. de Oliveira Tavares VD, Galvão-Coelho NL, Firth J, Rosenbaum S, Stubbs B, Smith L, et al. Reliability and Convergent Validity of Self-Reported Physical Activity Questionnaires for People With Mental Disorders: A Systematic Review and Meta-Analysis. J Phys Act Health. 2021 Jan 1;18(1):109–15.

9. Helmerhorst HHJ, Brage S, Warren J, Besson H, Ekelund U. A systematic review of reliability and objective criterion-related validity of physical activity questionnaires. International Journal of Behavioral Nutrition and Physical Activity. 2012 Dec 31;9(1):103.

---

## [Decision Letter · Decision Letter 1]

5 Feb 2025

Ethnic Variations in Cardiovascular Disease (CVD) Risk Factors and Associations with Prevalent CVD and CVD Mortality in the United States

PONE-D-24-43621R1

Dear Dr. Kuk,

We’re pleased to inform you that your manuscript has been judged scientifically suitable for publication and will be formally accepted for publication once it meets all outstanding technical requirements.

Kind regards,

Neftali Eduardo Antonio-Villa, MD PhD

Academic Editor

PLOS ONE

Additional Editor Comments (optional):

Reviewers' comments:

Reviewer's Responses to Questions

**Comments to the Author**

1. If the authors have adequately addressed your comments raised in a previous round of review and you feel that this manuscript is now acceptable for publication, you may indicate that here to bypass the “Comments to the Author” section, enter your conflict of interest statement in the “Confidential to Editor” section, and submit your "Accept" recommendation.

Reviewer #1: All comments have been addressed

2. Is the manuscript technically sound, and do the data support the conclusions?

Reviewer #1: Yes

3. Has the statistical analysis been performed appropriately and rigorously? 

Reviewer #1: Yes

4. Have the authors made all data underlying the findings in their manuscript fully available?

Reviewer #1: Yes

5. Is the manuscript presented in an intelligible fashion and written in standard English?

Reviewer #1: Yes

6. Review Comments to the Author

Reviewer #1: (No Response)

7. PLOS authors have the option to publish the peer review history of their article (what does this mean? ). If published, this will include your full peer review and any attached files.

**Do you want your identity to be public for this peer review?** For information about this choice, including consent withdrawal, please see our Privacy Policy .

Reviewer #1: **Yes: ** Afshin Heidari

---

## [Editor Report · Acceptance letter]

PONE-D-24-43621R1

PLOS ONE

Dear Dr. Kuk,

I'm pleased to inform you that your manuscript has been deemed suitable for publication in PLOS ONE. Congratulations! Your manuscript is now being handed over to our production team.

Kind regards,

on behalf of

Dr. Neftali Eduardo Antonio-Villa

Academic Editor

PLOS ONE